# Critical Roles of the Sphingolipid Metabolic Pathway in Liver Regeneration, Hepatocellular Carcinoma Progression and Therapy

**DOI:** 10.3390/cancers16050850

**Published:** 2024-02-20

**Authors:** Hiroyuki Nojima, Hiroaki Shimizu, Takashi Murakami, Kiyohiko Shuto, Keiji Koda

**Affiliations:** Department of Surgery, Teikyo University Chiba Medical Center, 3426-3, Anesaki, Ichihara, Chiba 299-0011, Japan; h-shimizu@med.teikyo-u.ac.jp (H.S.); gtrennsport3@gmail.com (T.M.); kshuto@med.teikyo-u.ac.jp (K.S.); k-koda@med.teikyo-u.ac.jp (K.K.)

**Keywords:** the sphingolipid metabolic pathway, liver regeneration, liver fibrosis, HCC

## Abstract

**Simple Summary:**

The sphingolipid metabolic pathway, an important signaling pathway, plays a crucial role in various physiological processes including immune regulation, cell proliferation, survival, and apoptosis. Recent studies have suggested that the metabolites from this pathway, such as ceramide, sphingosine 1-phosphate, and sphingosine 1-phosphate receptor, may have therapeutic potential for liver regeneration, fibrosis, and hepatocellular carcinoma (HCC). This review summarizes the mechanisms and therapeutic strategies targeting the sphingolipid metabolic pathway in liver regeneration, fibrosis, and HCC.

**Abstract:**

The sphingolipid metabolic pathway, an important signaling pathway, plays a crucial role in various physiological processes including cell proliferation, survival, apoptosis, and immune regulation. The liver has the unique ability to regenerate using bioactive lipid mediators involving multiple sphingolipids, including ceramide and sphingosine 1-phosphate (S1P). Dysregulation of the balance between sphingomyelin, ceramide, and S1P has been implicated in the regulation of liver regeneration and diseases, including liver fibrosis and hepatocellular carcinoma (HCC). Understanding and modulating this balance may have therapeutic implications for tumor proliferation, progression, and metastasis in HCC. For cancer therapy, several inhibitors and activators of sphingolipid signaling, including ABC294640, SKI-II, and FTY720, have been discussed. Here, we elucidate the critical roles of the sphingolipid pathway in the regulation of liver regeneration, fibrosis, and HCC. Regulation of sphingolipids and their corresponding enzymes may considerably influence new insights into therapies for various liver disorders and diseases.

## 1. Introduction

The sphingolipid metabolic pathway is an essential intracellular pathway consisting of biologically active sphingolipids, specifically sphingomyelin, ceramide, sphingosine, and sphingosine 1-phosphate (S1P), and their corresponding enzymes. The sphingolipid metabolic pathway plays a crucial role in various physiological processes including cell proliferation, survival, apoptosis, and immune regulation [1]. Maintaining a balance between various sphingolipid metabolites such as ceramide, sphingosine, and S1P is crucial for cellular homeostasis and various physiological processes [2]. Numerous studies have shown that the activation of signaling pathways such as HGF, EGFR, TGF-β, TNF-α, and inflammatory pathways (NF-κB, STAT3) promotes liver regeneration, and the activation of IL-6 and IL-8 signaling pathways may lead to tumor-associated inflammation and development of HCC [3,4,5,6]. In contrast to these signals, sphingolipids are not only essential components of cell membranes, but also function as autocrine and paracrine mediators that contribute to regeneration and HCC development [7,8,9]. Through cytokine-mediated events involved in liver regeneration, fibrosis and hepatocellular carcinoma are understood to a certain degree; however, how sphingolipids regulate these events is largely unknown [10,11]. Herein, we review the current knowledge on the role of sphingolipids in liver regeneration, fibrosis, HCC progression, and therapy.

## 2. Metabolism and Physiological Functions in Sphingolipids

The sphingolipid metabolic pathway is an essential intracellular pathway consisting of biologically active sphingolipids, specifically sphingomyelin, ceramide, sphingosine, and S1P, and their corresponding enzymes [12]. The balance between sphingomyelin, ceramide, and S1P in liver tissues is tightly regulated and is crucial for maintaining hepatic function [8]. Sphingomyelin can hydrolyze sphingomyelinases to produce ceramide, which can be further metabolized into sphingosine by ceramidases [13]. Ceramide is synthesized by several pathways, primarily the de novo synthesis, the salvage, and the sphingomyelin hydrolysis pathways. Ceramide synthases (CerSs) and sphingomyelinases (SMases) are the major enzymes responsible for ceramide production. CerSs generate ceramide from sphingosine, and SMases generate ceramides by hydrolyzing sphingomyelin [14]. S1P is produced by the phosphorylation of sphingosine by sphingosine kinases (SK1 and SK2). It can be dephosphorylated by specific phosphatases to regenerate sphingosine (Figure 1) [15]. The dephosphorylation of S1P can be catalyzed by several enzymes, including lipid phosphate phosphatases (LPPs) and protein phosphatases such as protein phosphatase 2A [16]. SK1 is mainly found in the lung and spleen and SK2 in the liver and heart; however, they are also widely distributed in other organs with overlapping expression [17]. With respect to sphingomyelin and ceramide balance, ceramide is often associated with pro-apoptotic and anti-proliferative effects. Sphingomyelin serves as a reservoir for ceramide generation and is involved in membrane structure [18]. In contrast, the balance between S1P and sphingosine levels is essential for regulating cellular responses such as cell survival, proliferation, migration, and immune cell trafficking [19]. Ceramide and S1P are involved in several cellular regulatory functions, and the relationship between the levels of these two sphingolipids plays an important role in regulating the balance between cell death and proliferation and in determining cell survival [20]. Imbalances in ceramide can have different effects on different cell types within the liver owing to differences in their metabolic requirements, functions, and responses to signaling molecules [21]. In hepatocytes, ceramide accumulation can disrupt mitochondrial function, leading to oxidative stress and death. In addition, ceramide-mediated inhibition of lipid synthesis pathways may affect hepatocyte function and contribute to liver dysfunction in liver regeneration. In hepatic stellate cells, ceramide-mediated signaling may stimulate the expression of fibrogenic cytokines and extracellular matrix proteins, leading to progression of liver fibrosis. Elevated levels of ceramide also activate inflammatory responses of Kupffer cells and promote the production of inflammatory cytokines and chemokines, contributing to liver inflammation and injury. In hepatic sinusoidal endothelial cells, ceramide-mediated endothelial dysfunction may contribute to liver injury, fibrosis, and impaired liver regeneration [21,22]. The sphingolipid pathway also plays a critical role in regulating lymphocyte function in the liver, contributing to both immune surveillance and immune-mediated liver disease (Figure 2) [23]. S1P gradients influence T cell trafficking in the liver. S1P1 regulates the trafficking of the immune system and the differentiation of effector and memory T cells [24]. Glycosphingolipids (GSLs) are involved in B-cell activation, antibody production, and immune surveillance in the liver [25]. Sphingomyelin metabolism such as sphingomyelinase inhibition modulates NK cell-mediated control of HCC [26]. Dysregulation of the sphingolipid pathway in lymphocytes may contribute to autoimmune hepatitis, a chronic inflammatory liver disease characterized by immune-mediated destruction of hepatocytes [27]. S1P functions as a signaling molecule by binding to specific S1P receptors (S1PR1–5) on the cell surface, activating various intracellular pathways [28]. As a second messenger, S1P functions independently of S1PR [29]. As a ligand and intracellular second messenger, S1P triggers various cellular responses, including proliferation, cell growth, survival, and angiogenesis. Among the S1PRs, S1PR1 and SIPR3 are associated with the activation of cell growth and migration, while SIPR2 has negative effects on cell growth and migration [30,31]. S1PR1 is also a positive regulator of EMT and tumor invasion (Figure 3) [32]. This balance is determined by the rates of synthesis, degradation, and S1PR-mediated signaling. Dysregulation of the balance between sphingomyelin, ceramide, and S1P has been implicated in liver regeneration and diseases, including liver fibrosis and HCC [8] (Table 1). Understanding and modulating this balance may have therapeutic implications for liver diseases. Targeting the sphingolipid metabolic pathway may have therapeutic potential for liver regeneration, fibrosis, and HCC.

## 3. Sphingolipid Metabolic Pathway in Liver Regeneration

Liver regeneration is a complex process that involves the proliferation of hepatocytes and the specific functions of numerous hormones, growth factors, and cytokines [60,61]. Post-hepatectomy liver failure (PHLF) continues to be a significant clinical challenge despite advances in hepatobiliary surgery over recent decades, and understanding the mechanism of liver regeneration after hepatectomy is essential to improving clinical outcomes [62,63]. Zabielski et al. reported that the activity of acidic sphingomyelinase increased gradually from the 4th to 24th hour after partial hepatectomy in rats, accompanied by a significant increase in ceramide content and reduction in S1P content [33]. Sun et al. reported that the increase in ceramide increased M1/M2 polarized bone-marrow-derived-macrophages to promote injury repair and regeneration in the liver after hepatectomy [34]. S1P, a bioactive sphingolipid metabolite, promotes liver regeneration by activating S1PRs on the hepatocytes. S1P has proliferative and anti-apoptotic effects and stimulates the production of IL-6 and VEGF in human liver sinusoidal endothelial cells (LSEC), which promotes the proliferation of hepatocytes. The results suggest that S1P promotes the production of IL-6 by human LSEC and that this effect enhances the proliferation of human hepatocytes, mainly through the activation of the STAT3 signaling pathway in a paracrine manner [64]. Serum and liver S1P concentrations were significantly increased in the associated liver partitioning and portal vein ligature for staged hepatectomy (ALPPS) groups after surgery in vivo [36]. The levels of ceramide and sphingosine increased, accompanied by increased levels of neutral sphingomyelinase localized in liver nuclei [65]. Albi et al. reported that sphingomyelinase is activated after hepatectomy with an acute decrease in sphingomyelin DNA synthesis, suggesting that nuclear sphingomyelin regulates cell proliferation [35]. Pro-inflammatory cytokines upregulate serine palmitoyl transferase activity in the liver, which plays an important role in initiating liver regeneration after partial hepatectomy [66].

Nojima et al. showed that exosomes released by hepatocytes fuse with and promote the proliferation of target hepatocytes, both in vitro and in vivo. Hepatocyte-derived exosomes deliver SK2 to target hepatocytes and induce hepatocyte proliferation via S1P generation. The proliferative effect of exosomes was prevented by ablating exosomal SK2 [37]. In addition, hepatocyte release of exosomes was dependent on CXCR1 and CXCR2, which regulate liver recovery and regeneration after I/R injury [67].

Similarly, Wu et al. reported that exosome-mimetic nanovesicles from hepatocytes promote hepatocyte proliferation in vitro and liver regeneration in vivo [38]. Nanovesicles promote hepatocyte proliferation and liver regeneration by significantly enhancing the SK2 content in recipient cells, similar to exosomes released from hepatocytes. Du et al. reported that exosomes produced by human-induced pluripotent stem cell-derived mesenchymal stromal cells could directly fuse with target hepatocytes and increase the activity of SK1 and synthesis of S1P, different in content from hepatocyte-derived exosomes. Human-induced pluripotent stem cell-derived mesenchymal stromal cells promote hepatocyte proliferation, both in vitro and in vivo [39]. In summary, the transfer of sphingolipids with exosomes or nanovesicles is an essential mechanism that can influence cell survival and proliferation during liver regeneration.

S1PRs are the key components of the sphingolipid metabolic pathway, and S1P signaling through S1PRs promotes cell survival and inhibits apoptosis [68], which play important roles in liver regeneration [37]. Wang et al. have reported that S1PR1 was selectively expressed at a high level in the blood vessels of the HCC tissue, compared to the normal tissue. In vitro and in vivo, high expression of S1PR1 in endothelial cells (ECs) promotes angiogenesis and HCC progression, suggesting that S1PR1 may be an important target for suppressing angiogenesis in HCC and that inhibiting S1PR1 may represent a promising approach toward HCC treatment [69]. Ikeda et al. have reported that S1P, mediated by S1PR2, regulates liver regeneration and fibrosis after liver injury [70]. Serriere-Lanneau revealed the role of S1PR2 in the wound healing response to acute liver injury via a mechanism involving enhanced proliferation of hepatic myofibroblasts [40]. Increased glucosylceramide suppresses cytokine- and growth factor-mediated signaling pathways in liver regeneration [71].

Given the known roles of various sphingolipids in pathways associated with hepatocyte proliferation, these studies provide important insights into the underlying mechanisms and may have a significant clinical impact on liver regeneration after hepatectomy.

## 4. Sphingolipid Metabolic Pathway in Liver Fibrosis and HCC Progression

Fibrotic liver injury is a progressive condition resulting from the inflammation of chronic hepatitis and can lead to portal hypertension, cirrhosis, and HCC progression [72,73]. Chronic liver congestion promotes liver fibrosis and HCC progression via S1P [74]. Neutrophil activation via S1PR2 plays an important role in the development of liver injury in the early stages of fatty liver [41].

Sphingolipids, such as S1P, SK, acidic sphingomyelinase, and sphingosine receptor (SR) have been implicated in various inflammatory liver diseases, such as steatosis and liver fibrosis [75,76]. Increased levels of S1P have been associated with the activation of hepatic stellate cells (HSCs), which are key players in liver fibrosis [77]. S1P promotes HSC activation, proliferation, and migration, and contributes to the deposition of extracellular matrix components, leading to fibrosis [68]. S1P levels in human fibrotic livers are significantly increased compared to those in normal livers [42,78]. In addition, the expressions of S1PR1 and S1PR3 in the liver are significantly increased in human fibrotic tissues [43]. S1PR1 and S1PR3 play critical roles in the angiogenic process required for fibrosis development and regulate the migration and fibrogenic activation of human hepatic stellate cells in S1P-stimulated liver fibrosis [63,64]. Hou et al. have reported that significantly increased gene expression of NLRP3 inflammasome components (NLRP3, pro-interleukin-1β, and pro-interleukin-18) and NLRP3 inflammasome activation were detected during chronic mouse liver injury. Treatment with S1PR2 siRNA significantly reduced NLRP3 inflammasome priming and activation, and attenuated liver inflammation and fibrosis [46]. S1PR2 and S1PR3 mediate bone marrow-derived monocyte/macrophage motility in cholestatic liver injury in mice [47]. SK1 upregulation is involved in TGF-beta1-induced activation of HSCs during liver fibrogenesis [48]. Ceramide, another important sphingolipid, has both pro-apoptotic and pro-inflammatory effects [2]. Ceramide, which acts synergistically with S1P, also plays a role in liver fibrosis and cirrhosis [79].

HCC is the most common type of liver cancer and is characterized by uncontrolled cell proliferation and survival [4]. Recent studies have suggested that dysregulation of sphingolipid metabolism may contribute to the development and progression of liver cancers including HCC [7,49]. Numerous sphingolipids regulate different aspects of the development and progression of cancer as well as the response to anti-cancer therapy [80]. S1P regulates resistance to proliferation, inflammation, angiogenesis, and apoptosis. S1P induced the proportion of cells in S phase in HCC cells, whereas S1P decreased the proportion of cells in both early and late apoptosis [81]. Liver S1P levels were positively correlated with fibrosis and tumor development, suggesting that LPS and S1P play key roles in the pathogenesis of congestive hepatopathy [74]. In contrast, ceramides are involved in regulating cancer cell proliferation, differentiation, senescence, and apoptosis [82]. Significantly increased serum concentrations of several long- and very-long-chain (C16-C24) ceramides were observed in patients with HCC compared to cirrhotic patients [83]. Li et al. showed that the level of sphingomyelin in the plasma membrane level is a key factor in regulating hepatocyte tumorigenesis [84].

Regarding S1PR expression in HCC, a study has shown that the sphingolipid metabolic pathway is upregulated, leading to the promotion of vascular invasion and EMT by activating the S1PR1 in hepatocytes [32]. In patients with HCC, the high S1PR1 expression group had a significantly shorter overall survival than the low expression group. In addition, high S1PR1 expression was significantly associated with shorter relapse-free survival, increased risk of portal and hepatic venous invasion, and intrahepatic metastasis. S1PR1 overexpression were positively correlated with the expression of vimentin and MMP-9, which contribute to the initiation of EMT. EMT-induced vascular invasion exhibits enhanced cancer stem cell properties, establishes a metastatic niche, increases the capacity for hematogenous metastasis, and is associated with poor outcome in patients with HCC (Figure 4) [32]. Ji et al. showed that S1PR1 and angiotensin II overexpression was associated with HCC progression, indicating that S1P/S1PR may serve as valuable prognostic biomarkers for HCC [50]. S1PR1 also induces metabolic reprogramming of ceramide in vascular endothelial cells, affecting angiogenesis and progression in HCC [69].

Regarding sphingosine kinase in HCC, SK1 is involved in the development and progression of hepatocellular carcinoma (HCC) cells [85,86]. Simultaneous high expression of SK1 and ABCC1 transporter, which reflects S1P export, was associated with the enhancement of HCC progression [9]. In patients with HCC, high SK1 expression is correlated with shorter overall survival and SK1 has been implicated in the resistance of HCC to oxaliplatin [51]. In vivo, SK1 depletion inhibits liver tumor formation in diethylnitrosamine-treated mice [87]. Highly upregulated in liver cancer (HCLC), SK1, which is involved in tumor angiogenesis, is elevated in HCC cells, and the silencing of SK1 remarkably abolished HCLC-enhanced tumor angiogenesis in vitro and in vivo [88].

## 5. Sphingolipid Metabolic Pathway in HCC Therapy

Systemic therapy with multiple targeted tyrosine kinase inhibitors (TKIs), including sorafenib, lenvatinib, regorafenib, and cabozantinib as well as, anti-PD-1 and anti-CTLA-4 immunotherapy, has been widely used to treat HCC (Table 2) [54,89,90,91,92,93,94,95,96,97,98,99,100]. Targeting the metabolites of sphingolipid metabolic pathways, such as S1PR, is also a potential therapeutic strategy for HCC [101]. Several S1PR1 modulators, including FTY720, a synthetic analog of S1P, sensitize HCC cells to cytotoxic effects induced by treatment with sorafenib [52]. In contrast, CYM5520, a selective S1PR2 agonist, significantly suppressed the migration of HuH7 cells induced by HGF, suggesting that S1P inhibits HCC cell migration through S1PR2 [58]. These agents may have yielded promising results in preclinical studies of HCC and liver regeneration, and further research is needed to determine the safety and efficacy of S1PR modulators in clinical settings. Regarding ceramide in HCC therapy, Fenretinide, which reduces ceramide synthesis, inhibits proliferation and migration of human hepatocellular carcinoma HepG2 cells through the p38 MAPK pathway [102]. The loading of short-chain C6 ceramide onto polyethylene glycol and polyethyleneimine-conjugated nano-sized GOs can also result in high cancer cell killing potential in HCC [23]. Autophagy dysfunction is associated with synergistic combination therapy of nanoliposomal C6-ceramide and vinblastine in HCC [103]. Inhibition of alkaline ceramidase 3, which hydrolyzes long-chain unsaturated ceramides to produce free fatty acids and sphingosine, results in intracellular exhaustion of S1P and inhibits activation of S1PR2/PI3K/AKT signaling in HCC cells [59]. Furthermore, with respect to SK in HCC therapy, the antiproliferative effect of quercetin on HepG2 hepatocarcinoma cells was enhanced by combination with fingolimod owing to the downregulation of SK1 [55]. The dual sphingosine kinase inhibitor SKI II increases sensitivity to 5-fluorouracil in HepG2 via suppression of osteopontin and FAK/IGF-1R signaling [104]. Cheng et al. showed that cinobufotalin, which inhibits SK1, induces growth inhibition and apoptosis in HCC cells by producing ceramide [105]. Downregulation of ceramide transfer protein by SK2 ablation also suppresses fatty liver-associated hepatocellular carcinoma [53]. SK2 mediated the phosphorylation of FTY720, resulting in metabolic inactivation of its anti-tumor activity. SK2 RNAi nanoparticles also suppress tumors by downregulating exosomal microRNA in cancer cells [106]. The targeting of SK2 by ABC294640 potently reduces the resistance of HCC cells to regorafenib both in vitro and in vivo through NF-κB and STAT3 activation [56]. Dual inhibition of the SK1 and Ras/ERK pathways resulted in enhanced inhibition of human hepatoma cell growth [107]. Fumonisin B1 and SKI II were relatively effective in inhibiting cell proliferation in HepG2 and Huh7.5 cells [108]. S1P lyase facilitates cancer progression by converting S1P into glycerophospholipids [57].

## 6. Conclusions

Sphingolipids are important regulators of liver homeostasis and oncogenesis, suggesting that they may play important roles in the regulation of liver regeneration and liver cancer progression. Novel therapies for liver regeneration or control of HCC progression may have significant impact through the regulation of sphingolipids and their enzymes. In conclusion, targeting the metabolites of sphingolipid metabolic pathways, such as S1PR, ceramide, SK, and S1P, may present a promising technique for treating HCC and promoting liver regeneration. Sphingolipid-targeting drugs for the treatment of HCC have been tested and have yielded promising results. Further research is needed to fully understand the mechanisms involved in this pathway and develop effective therapeutic strategies for liver disease.

## Figures and Tables

**Figure 1 cancers-16-00850-f001:**
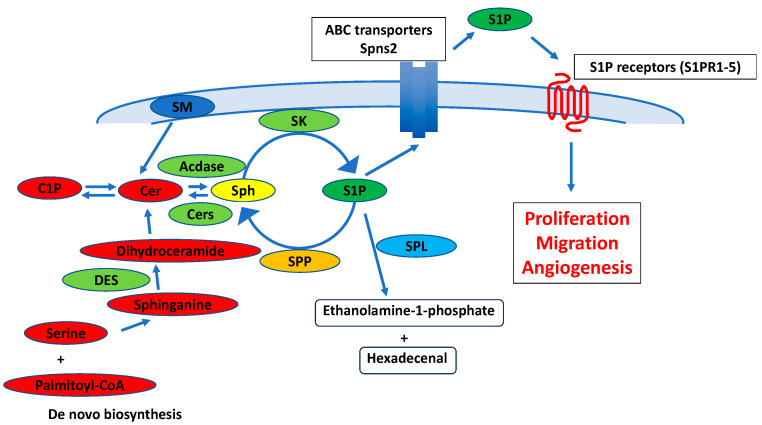
Sphingolipid metabolic pathway in liver regeneration and HCC progression. Numerous agonists, including cytokines and growth factors, activate cytosolic SK to translocate to the plasma membrane and produce S1P. S1P can be irreversibly degraded by S1P lyase (SPL). It can also be dephosphorylated by S1P phosphatases (SPP). After being secreted by ABC transporters or Spns2, S1P binds and activates S1P receptors (S1PR1-5) and regulates numerous cellular functions such as proliferation, angiogenesis and migration. In de novo sphingolipid biosynthesis, serine and palmitoyl-CoA are converted to sphinganine and further modified to form ceramide. SM: sphingomyelin, Cer: ceramide, Cers: ceramide synthtase, ACdase: acid ceramidase, Sph: sphingosine, SK: sphingosine kinase, SPP: sphingosine phosphatase, SPL: sphingosine lyase, DES: dihydroceramide desaturase.

**Figure 2 cancers-16-00850-f002:**
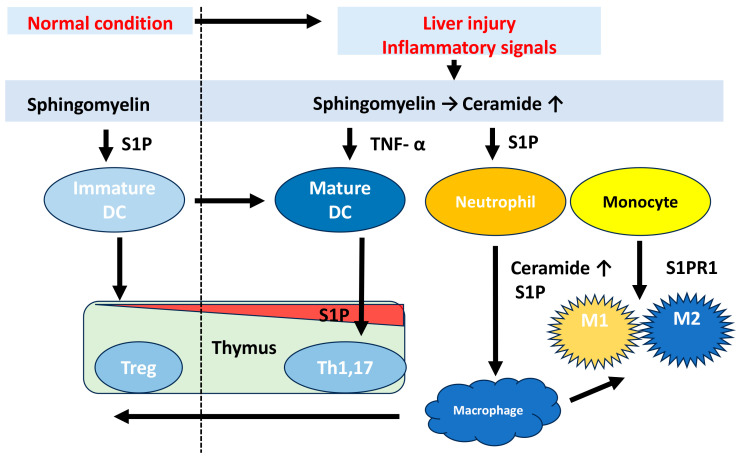
Immunological functions of sphingolipids. Increased intracellular and extracellular sphingolipids, such as S1P or ceramide, activate the immune response. The S1P/S1PR system plays a key role in neutrophil recruitment and monocyte activation. S1P also directly modulates functional activity of dendritic cells. Macrophage efferocytosis is a key player in the resolution of inflammation. DC: Dendritic cells, Treg: regulatory T cells.

**Figure 3 cancers-16-00850-f003:**
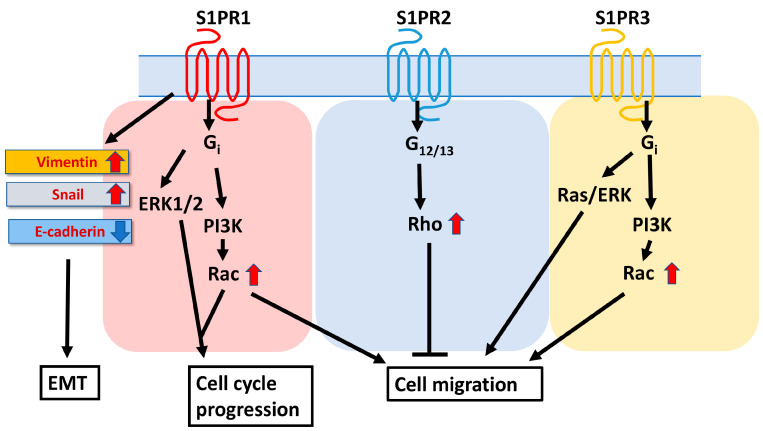
S1PR signaling pathways in liver regeneration and HCC progression. Sphingosine-1-phosphate (S1P) acts as a lipid mediator of various cellular responses such as cell cycle, proliferation, and migration, and acts primarily through S1P receptors (S1PR1 to S1PR3). S1PR1 mediates cell migration via the Gi/Rac pathway, whereas S1PR2 mediates inhibition of migration via the G12/13/Rho pathway. SIPR3 also promotes cell migration via the Gi/Rac or Ras/ERK pathway. S1PR1 overexpression was positively correlated with vimentin and MMP-9 expression and negatively correlated with E-cadherin expression. In addition, S1PR1 overexpression induced EMT and enhanced tumor invasion and cancer stemness.

**Figure 4 cancers-16-00850-f004:**
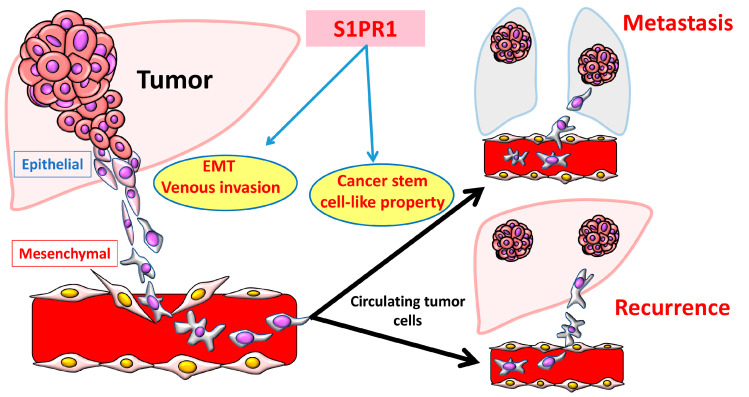
S1PR1 expression via EMT-induced vascular invasion and increased cancer stem cell properties in HCC metastasis and recurrence. S1PR1 overexpression, via EMT-induced vascular invasion and enhanced cancer stem cell properties, establishes a metastatic niche and increases the capacity for hematogenous metastasis as well as recurrence.

**Table 1 cancers-16-00850-t001:** Characteristics of sphingolipids in liver regeneration, fibrosis, and HCC.

Models	Sphingolipids	Pharmalogical Approach		References
Hepatectomy	Ceramide ↑, ASM ↑		Liver regenerarion ↑	[33]
Hepatectomy	Ceramide ↑, S1P ↑		Liver regenerarion ↑	[34]
Hepatectomy	sphingomyelinase ↑		Liver regenerarion ↑	[35]
Hepatectomy (ALPPS)	S1P (serum and liver) ↑			[36]
Hepatectomy	SK2 ↑, S1P ↑	Hepatocyte-derived exosomes	Liver regenerarion ↑	[37]
Hepatectomy	SK2 ↑	Nanovesicles from hepatocytes	Liver regenerarion ↑	[38]
Ischemia reperfusion	SK1 ↑, S1P ↑	Human-induced pluripotent stem cell-derived mesenchymal stromal cells	Liver regenerarion ↑	[39]
Acute liver injury	S1PR2 ↑		Liver regenerarion ↑	[40]
Liver fibrosis		S1PR2 inhibition	Liver regenerarion ↑ fibrosis ↓	[41]
Liver fibrosis	S1P ↑		fibrosis ↑	[42]
Liver fibrosis	S1PR1 ↑, S1PR3 ↑,		fibrosis ↑	[43,44]
Liver fibrosis	S1PR1 ↑, S1PR3 ↑, S1P ↑		fibrosis ↑	[45]
Liver fibrosis		S1PR2 inhibition	fibrosis ↓	[46]
Liver fibrosis	S1PR2 ↑, S1PR3 ↑,		fibrosis ↑	[47]
Liver fibrosis	SK1 ↑		fibrosis ↑	[48]
HCC	Ceramide ↑, S1P ↑, SK1 ↑			[49]
HCC	S1PR1↑	S1PR1 inhibition	HCC Progression ↑	[32]
HCC	S1PR1 ↑		HCC Progression ↑	[50]
HCC	S1PR1 ↑, Ceramide ↑		HCC Progression ↑	[51]
HCC		FTY720	Sorafenib-mediated cytotoxicity	[52]
HCC	SK2 ↑		HCC Progression ↓	[53]
HCC	SK1 ↓	Cinobufotalin	HCC cells ↓	[54]
HCC	SK1 ↓	Fingolimod	HCC cells ↓	[55]
HCC	SK2 ↓	ABC294640	HCC cells ↓	[56]
HCC	S1P lyase ↑		HCC Progression ↑	[57]
HCC	S1PR2 ↑	CYM5520	HCC Progression ↓	[58]
HCC	S1PR2 ↑	Alkaline ceramidase 3 inhibition	HCC Progression ↓	[59]
	↑ up-regulation		↑ up-regulation	
	↓ down-regulation		↓ down-regulation	

**Table 2 cancers-16-00850-t002:** Characteristics of drugs in HCC therapy.

References	Trials	Agent	Targets	Indication	Overall Survival (Months)
[70]	SHARP	Sorafenib	VEGFR, c-KIT, PDGFR, RET and Ras/Raf/MEK/ERK	First-line	10.7
[40]	Asia–Pacific	Lenvatinib	VEGFR, PDGFR, FGFR, KIT and RET	First-line	6.5
[71]	REFLECT	Bevacizumab Atezolizumab	VEGF and PD-L1	First-line	13.6
[72]	IMbrave150	Durvalumab Tremelimumab	PDL-1 and CTLA-4	First-line	16.4
[73]	HIMALAYA	Atezolizumab Cabozantinib	PD-L1 and VEGF	First-line	15.4
[74]	RESORCE	Regorafenib	VEGFR, FGFR, PDGFR, B-RAF, RET and KIT	Second	10.6
[41]	KEYNOTE 224	Pembrolizumab	PD-1	Second	12.9
[75]	KEYNOTE 240	Pembrolizumab	PD-1	Second	13.9
[76]	CELESTIAL	Cabozantinib	VEGFR, AXL, c-MET, KIT and RET	Second	10.2
[77]	REACH-2	Ramucirumab	VEGFR2	Second	8.5
[78]	CHECKMATE 040	Nivolumab and Ipilimumab	PD-1 and CTLA-4	Second	arm A: 22.8 B: 12.5 C: 12.7
[42]	KEYNOTE 394	Pembrolizumab	PD-1	Second	14.6

## Data Availability

Data will be made available by the corresponding author upon request.

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
