# Peer review of "Critical Roles of the Sphingolipid Metabolic Pathway in Liver Regeneration, Hepatocellular Carcinoma Progression and Therapy"

_cancers, 2024, doi:10.3390/cancers16050850_

Round 1

Reviewer 1 Report

Comments and Suggestions for Authors

Comments and suggestions to authors

In this article, Nojima et al. reviewed the critical roles of the sphingolipid metabolic pathway, a regulatory pathway involving immunity, proliferation survival, and apoptosis. They further discussed several metabolites regulating liver regeneration fibrosis, HCC, and therapeutic strategies.

While certain parts may need additional revisions, the manuscript has the potential to be of interest to a large audience and readers and publication in the journal.

 1.     Please improve the language throughout the manuscript. Please define the abbreviated terms when they first appear and later use the abbreviations throughout the manuscript.

2.     There are many lengthy statements in the manuscript. Please shorten them.

3.     The word length of the main text is not according to the required limit for the journal, which is quite concerning and should be comprehensively revised in the manuscript and enriched with more recent literature with added details in all sections.

4.     Moreover, adding a comprehensive section on cancer therapy in the manuscript will potentially increase the readers' interest in exploiting the pathway in hepatocellular carcinoma and therapy. Also, discuss the enzymes, mediators, and components in the sphingolipid pathway for their role in hepatocellular carcinoma anticancer therapy with recent literature.

5.     Likewise, please provide a detailed therapeutic table consisting of FDA-approved, and drugs in clinical trials.

6.     Subsequently, change the title accordingly after adding the anticancer therapy section.

7.     Please use biorender for all the figures and provide them in at least 300 dpi resolution. Also, use consistent fonts in all figures.

8.     Please briefly describe what is occurring in the figure captions (pathways, components, and vice versa).

9.     Please revise the abstract precisely to the literature presented.

10.  Authors have mentioned roles in immune regulation but presented no literature on it in the main text, and this pathway is crucial in immune regulation, homeostasis, and innate immunity.

11.  Also, provide a figure about the roles of sphingolipids in immune regulation and cancer immunity.

12.  Correspondingly, discuss the roles of the sphingolipid pathway and their components in lymphocytes.

13.  The introduction section is concise and abridged; please revise it, introducing the topic and the contents discussed; what are the gaps? Read relevant literature and see how others introduced the topics to understand it better.

14.  L42-L47, please revise and remove the repetitions. Also, remove repetitions throughout the manuscript.

15.  L49, please change, Sphingomyelin can be hydrolyzed by sphingomyelinases “to” Sphingomyelin can hydrolyze sphingomyelinases

16.  L70, please change, S1PR1 also serves as a “to” S1PR1 is also a

17.  L143, please change, contributes to deposition “to” contributes to the deposition

18.  Figure 1: Please move the abbreviation of the terms from the figure to the end of the figure caption.

19.  Figure 2: inconsistent font sizes and arrows. Please revise.

20.  Figure 3: Poor illustration, inconsistent font sizes and overlaps. Please revise.

21.  Please make sure to revise conclusions after changes.

22.  Please update the reference with recent literature.

Comments on the Quality of English Language

1.     Please improve the language throughout the manuscript. Please define the abbreviated terms when they first appear and later use the abbreviations throughout the manuscript.

2.     There are many lengthy statements in the manuscript. Please shorten them.

15.  L49, please change, Sphingomyelin can be hydrolyzed by sphingomyelinases “to” Sphingomyelin can hydrolyze sphingomyelinases

16.  L70, please change, S1PR1 also serves as a “to” S1PR1 is also a

17.  L143, please change, contributes to deposition “to” contributes to the deposition

Author Response

Reviewers' comments:

Reviewer #1: In this article, Nojima et al. reviewed the critical roles of the sphingolipid metabolic pathway, a regulatory pathway involving immunity, proliferation survival, and apoptosis. They further discussed several metabolites regulating liver regeneration fibrosis, HCC, and therapeutic strategies. 

While certain parts may need additional revisions, the manuscript has the potential to be of interest to a large audience and readers and publication in the journal.

  1. Please improve the language throughout the manuscript. Please define the abbreviated terms when they first appear and later use the abbreviations throughout the manuscript.

Response to comment 1:

Thank you for your suggestion. As you point out, we have defined the abbreviated terms at the time of their first appearance and later use the abbreviations throughout the manuscript.

  1. There are many lengthy statements in the manuscript. Please shorten them.

Response to comment 2:

Thank you for your suggestion. As you point out, we have shortened many lengthy statements.

  1. The word length of the main text is not according to the required limit for the journal, which is quite concerning and should be comprehensively revised in the manuscript and enriched with more recent literature with added details in all sections.

Response to comment 3:

Thank you for your suggestion. As you point out, we have revised the manuscript, enriched it with more recent literature, and added more details in all sections.

4.Moreover, adding a comprehensive section on cancer therapy in the manuscript will potentially increase the readers' interest in exploiting the pathway in hepatocellular carcinoma and therapy. Also, discuss the enzymes, mediators, and components in the sphingolipid pathway for their role in hepatocellular carcinoma anticancer therapy with recent literature. 

Response to comment 4:

Thank you for your suggestion. As you point out, we have revised the manuscript, and discussed the enzymes, mediators, and components in the sphingolipid pathway for their role in hepatocellular carcinoma anticancer therapy with recent literature. 

  1. Likewise, please provide a detailed therapeutic table consisting of FDA-approved, and drugs in clinical trials. 

Response to concern 5:

Thank you for your suggestions. As you suggest, we have created the table of FDA-approved, and drugs in clinical trials and summarized the characteristics of each case.

  1. Subsequently, change the title accordingly after adding the anticancer therapy section.

Response to concern 6:

Thank you for your suggestions. As you suggest, we have created the anticancer therapy section.

  1. Please use biorender for all the figures and provide them in at least 300 dpi resolution. Also, use consistent fonts in all figures.

Response to comment 7:

Thank you for your suggestions. As you suggest, we have created all the figures and provide them in at least 300 dpi resolution with consistent fonts.

  1. Please briefly describe what is occurring in the figure captions (pathways, components, and vice versa).

Response to comment 8:

Thank you for your suggestions. As you suggest, we have described all the figures.

  1. Please revise the abstract precisely to the literature presented.

Response to comment 9:

Thank you for your suggestion. As you point out, we have revised the abstract precisely to the literature presented.

  1. Authors have mentioned roles in immune regulation but presented no literature on it in the main text, and this pathway is crucial in immune regulation, homeostasis, and innate immunity.

Response to comment 10:

Thank you for your suggestion. As you point out, we have added the immune regulation, homeostasis, and innate immunity in the sentence.

  1. Also, provide a figure about the roles of sphingolipids in immune regulation and cancer immunity.

Response to comment 11:

Thank you for your suggestion. As you point out, we have added the figure about the roles of sphingolipids in immune regulation and cancer immunity.

  1. Correspondingly, discuss the roles of the sphingolipid pathway and their components in lymphocytes.

Response to comment 12:

Thank you for your suggestion. As you point out, we have added the roles of the sphingolipid pathway and their components in lymphocytes in the sentence.

  1. The introduction section is concise and abridged; please revise it, introducing the topic and the contents discussed; what are the gaps? Read relevant literature and see how others introduced the topics to understand it better.

Response to comment 13:

Thank you for your suggestion. As you point out, we have revised the sentence in introduction.

  1. L42-L47, please revise and remove the repetitions. Also, remove repetitions throughout the manuscript.

Response to comment 14:

Thank you for your suggestion. As you point out, we have revised and remove the repetitions.

  1. L49, please change, Sphingomyelin can be hydrolyzed by sphingomyelinases “to” Sphingomyelin can hydrolyze sphingomyelinases

Response to comment 15:

Thank you for your suggestion. As you point out, we have revised the sentence.

  1. L70, please change, S1PR1 also serves as a “to” S1PR1 is also a

Response to comment 16:

Thank you for your suggestion. As you point out, we have revised the sentence.

  1. L143, please change, contributes to deposition “to” contributes to the deposition

Response to comment 17:

Thank you for your suggestion. As you point out, we have revised the sentence.

  1. Figure 1: Please move the abbreviation of the terms from the figure to the end of the figure caption. 

Response to comment 18:

Thank you for your suggestion. As you point out, we have moved abbreviation of the terms from the figure to the end of the figure caption. 

  1. Figure 2: inconsistent font sizes and arrows. Please revise. 

Response to comment 19:

Thank you for your suggestion. As you point out, we have corrected inconsistent font sizes and arrows in Figure 2.

  1. Figure 3: Poor illustration, inconsistent font sizes and overlaps. Please revise. 

Response to comment 20:

Thank you for your suggestion. As you point out, we have created the Figure 3 according to your suggestion.

  1. Please make sure to revise conclusions after changes.

Response to comment 21:

Thank you for your suggestion. As you suggest, we have revised conclusions.

  1. Please update the reference with recent literature.

Response to comment 22:

Thank you for your suggestion. As you point out, we have updated the references with recent literature.

Reviewer 2 Report

Comments and Suggestions for Authors

In this manuscript the authors review the therapeutic importance of intracellular signaling pathway sphingolipid metabolic pathway in liver regeneration, fibrosis and HCC.  It is a good manuscript and will contribute to future research to the community working on liver. However, I suggest to improve the manuscript content since the same context is repeated 2-3 times and so please accept after minor revision.

My comments are as follows:

1.     Please improve the introduction with better literature survey – example give a short review of other important pathways that generally contribute to liver regeneration, hcc. How do they contribute and how is sphingolipid metabolic pathway different ?

2.     Please give details from literature on what phosphatases dephosphorylate S1P

3.     Lines 67-70: repeats the same thing twice

4.     How would the imbalance of metabolite affect different cell types in liver – eg: parenchymal, non-parenchymal?

5.     Fig 1: annotate S1P, include downstream function of ceramide

6.     Mechanistic modeling approach that incorporates different mechanism of action (eg: PMID: 25152891, PMID: 30651095, PMID: 28052241,) can help verify if this pathway could contribute to liver regeneration, fibrosis and hcc prior to performing experiments. Or experimental data from such experiments can be used to build mechanistic models of liver regeneration, hcc such as eg: PMID: 28615926, PMID: 36634921 to make further therapeutic impact of the balance of metabolites in sphingolipid metabolic pathway on liver regeneration,fibrosis.  

Author Response

Reviewer #2: In this manuscript the authors review the therapeutic importance of intracellular signaling pathway sphingolipid metabolic pathway in liver regeneration, fibrosis and HCC.  It is a good manuscript and will contribute to future research to the community working on liver. However, I suggest to improve the manuscript content since the same context is repeated 2-3 times and so please accept after minor revision. 

My comments are as follows:

  1. Please improve the introduction with better literature survey – example give a short review of other important pathways that generally contribute to liver regeneration, hcc. How do they contribute and how is sphingolipid metabolic pathway different ?

Response to comment 1:

Thank you for your suggestion. As you point out, we added other important pathways that generally contribute to liver regeneration, HCC. We also added the contribution and differences of other pathways.

  1. Please give details from literature on what phosphatases dephosphorylate S1P

Response to comment 2:

Thanks for your suggestion. As you point out, we have added a description of which phosphatases dephosphorylate S1P.

  1. Lines 67-70: repeats the same thing twice

Response to comment 3:

Thank you for your suggestion. As you point out, we have revised and remove the repetitions.

  1. How would the imbalance of metabolite affect different cell types in liver – eg: parenchymal, non-parenchymal?

Response to comment 4:

Thank you for your suggestion. As you point out, we added details about the imbalance of ceramide mediated affect in hepatic different cells.

  1. Fig 1: annotate S1P, include downstream function of ceramide

Response to comment 5:

Thank you for your suggestion. As you point out, we have include downstream function of ceramide in Fig 1.

  1. Mechanistic modeling approach that incorporates different mechanism of action (eg: PMID: 25152891, PMID: 30651095, PMID: 28052241,) can help verify if this pathway could contribute to liver regeneration, fibrosis and hcc prior to performing experiments. Or experimental data from such experiments can be used to build mechanistic models of liver regeneration, hcc such as eg: PMID: 28615926, PMID: 36634921 to make further therapeutic impact of the balance of metabolites in sphingolipid metabolic pathway on liver regeneration,fibrosis.

Response to comment 6:

Thank you for your suggestion. As you point out, we added couple of sentences as well as references.

Reviewer 3 Report

Comments and Suggestions for Authors

The manuscript titled "Critical Roles of the Sphingolipid Metabolic Pathway in Liver Regeneration and Hepatocellular Carcinoma Progression" provides an in-depth analysis of the role of sphingolipid metabolism in liver regeneration and the progression of hepatocellular carcinoma. The manuscript is well-organized and worth publishing.

Several notes:

1 Please revise the text:

There should be a consistent space between word and reference numbers throughout the entire text.

2.  Please provide detailed figure legends for all figures. This will help readers better understand the content of the figures and improve the overall clarity of the manuscript.

3. Line 92-93. Please clarify the text. It is unclear how Il-6 and VEGF play a role in proliferation or apoptosis at the same time.

Line 95-96: Please check the English grammar in this sentence.

Line 104-115:  Nojima, Wu, and Du et al. have made similar findings; please describe the similar and differing points.

Lines 146-149: Please combine two sentences in one sentence.

Line 151: It's unclear how NLRP3 inflammation and cell death relate to the text. Please add additional information.

Line 161: S1P regulates resistance to proliferation and apoptosis. Clarify the text.

Line 200: Please include an abbreviation for hepatocellular carcinoma cells.

Comments on the Quality of English Language

The manuscript titled "Critical Roles of the Sphingolipid Metabolic Pathway in Liver Regeneration and Hepatocellular Carcinoma Progression" provides an in-depth analysis of the role of sphingolipid metabolism in liver regeneration and the progression of hepatocellular carcinoma. The manuscript is well-organized and worth publishing.

Several notes:

1 Please revise the text:

There should be a consistent space between word and reference numbers throughout the entire text.

2.  Please provide detailed figure legends for all figures. This will help readers better understand the content of the figures and improve the overall clarity of the manuscript.

3. Line 92-93. Please clarify the text. It is unclear how Il-6 and VEGF play a role in proliferation or apoptosis at the same time.

Line 95-96: Please check the English grammar in this sentence.

Line 104-115:  Nojima, Wu, and Du et al. have made similar findings; please describe the similar and differing points.

Lines 146-149: Please combine two sentences in one sentence.

Line 151: It's unclear how NLRP3 inflammation and cell death relate to the text. Please add additional information.

Line 161: S1P regulates resistance to proliferation and apoptosis. Clarify the text.

Line 200: Please include an abbreviation for hepatocellular carcinoma cells.

Author Response

Reviewer #3: The manuscript titled "Critical Roles of the Sphingolipid Metabolic Pathway in Liver Regeneration and Hepatocellular Carcinoma Progression" provides an in-depth analysis of the role of sphingolipid metabolism in liver regeneration and the progression of hepatocellular carcinoma. The manuscript is well-organized and worth publishing.

Several notes:

1 Please revise the text: 

There should be a consistent space between word and reference numbers throughout the entire text.

Response to comment 1:

Thank you for your suggestion. As you point out, we spaced between word and reference numbers throughout the entire text.

  1. Please provide detailed figure legends for all figures. This will help readers better understand the content of the figures and improve the overall clarity of the manuscript.

Response to comment 2:

Thank you for your suggestion. As you point out, we have added detailed figure legends for all figures. 

  1. Line 92-93. Please clarify the text. It is unclear how Il-6 and VEGF play a role in proliferation or apoptosis at the same time.

Response to comment 3:

Thank you for your suggestion. As you point out, we have added detailed about Il-6 and VEGF playing a role in proliferation or apoptosis at the same time.

Line 95-96: Please check the English grammar in this sentence.

Response to comment 4:

Thank you for your suggestion. As you point out, we have check the English grammar and revised this sentence.

Line 104-115:  Nojima, Wu, and Du et al. have made similar findings; please describe the similar and differing points.

Response to comment 5:

Thank you for your suggestion. As you point out, we have added the similar and differing points between exosome reports.

Lines 146-149: Please combine two sentences in one sentence.

 Response to comment 6:

Thank you for your suggestion. As you point out, we have combined two sentences in one sentence.

Line 151: It's unclear how NLRP3 inflammation and cell death relate to the text. Please add additional information.

Response to comment 7:

Thank you for your suggestion. As you point out, we have revised the sentence.

Line 161: S1P regulates resistance to proliferation and apoptosis. Clarify the text.

Response to comment 8:

Thank you for your suggestion. As you point out, we have revised the sentence.

Line 200: Please include an abbreviation for hepatocellular carcinoma cells.

Response to comment 9:

Thank you for your suggestion. As you point out, we have added HepG2 in the sentence.

Round 2

Reviewer 1 Report

Comments and Suggestions for Authors

The revised version of the manuscript has been sufficiently improved to warrant publication in Cancers.

Reviewer 3 Report

Comments and Suggestions for Authors

NBA

Comments on the Quality of English Language

NA